# Osteoporosis Pre-Screening Using Ensemble Machine Learning in Postmenopausal Korean Women

**DOI:** 10.3390/healthcare10061107

**Published:** 2022-06-14

**Authors:** Youngihn Kwon, Juyeon Lee, Joo Hee Park, Yoo Mee Kim, Se Hwa Kim, Young Jun Won, Hyung-Yong Kim

**Affiliations:** 1Insilicogen, Inc., Yongin-si 16954, Korea; yikwon@insilicogen.com; 2AIDX, Inc., Yongin-si 16954, Korea; jylee@aidx.kr (J.L.); jheepark@aidx.kr (J.H.P.); 3Department of Internal Medicine, International St. Mary’s Hospital, Catholic Kwandong University College of Medicine, Incheon 22711, Korea; ymkim@ish.ac.kr (Y.M.K.); endojune@ish.ac.kr (S.H.K.)

**Keywords:** machine learning, feature selection, osteoporosis, postmenopausal women, pre-screening, risk assessment

## Abstract

As osteoporosis is a degenerative disease related to postmenopausal aging, early diagnosis is vital. This study used data from the Korea National Health and Nutrition Examination Surveys to predict a patient’s risk of osteoporosis using machine learning algorithms. Data from 1431 postmenopausal women aged 40–69 years were used, including 20 features affecting osteoporosis, chosen by feature importance and recursive feature elimination. Random Forest (RF), AdaBoost, and Gradient Boosting (GBM) machine learning algorithms were each used to train three models: A, checkup features; B, survey features; and C, both checkup and survey features, respectively. Of the three models, Model C generated the best outcomes with an accuracy of 0.832 for RF, 0.849 for AdaBoost, and 0.829 for GBM. Its area under the receiver operating characteristic curve (AUROC) was 0.919 for RF, 0.921 for AdaBoost, and 0.908 for GBM. By utilizing multiple feature selection methods, the ensemble models of this study achieved excellent results with an AUROC score of 0.921 with AdaBoost, which is 0.1–0.2 higher than those of the best performing models from recent studies. Our model can be further improved as a practical medical tool for the early diagnosis of osteoporosis after menopause.

## 1. Introduction

Osteoporosis is a representative disease that accompanies aging and is closely related to skeletal fractures and deaths [1]. Therefore, a methodology for early diagnosis and prevention has been proposed. Osteoporosis is diagnosed by measuring bone mineral density (BMD) using dual-energy X-ray absorptiometry (DXA) equipment [2]. However, the associated costs are expensive [3]. Hence, with the accelerating growth of aged populations, the financial burdens of individuals and governments are increasing dramatically [4,5]. Notably, a pre-screening diagnosis method that leverages data from surveys and checkups to evaluate osteoporosis risk in advance would greatly benefit prevention and treatment while reducing economic and financial burdens on society. For these reasons, pre-screening diagnosis methods have been actively studied. Thus, many conventional methods of predicting osteoporosis risk are used, including the Osteoporosis Self-Assessment Tool for Asians [6], the osteoporosis risk assessment instrument [7], simple calculated osteoporosis risk estimation [8], and the osteoporosis index of risk [9,10], because these methods rely on only two or three features to predict osteoporosis simply. However, because enormous amounts of medical data are collected nowadays, it is necessary to apply complicated statistical methods to utilize data in advance for better results [3].

Machine learning is an artificial intelligence technique for learning patterns and predicting outcomes based on input data [11]. Machine learning is especially effective at identifying trends and making predictions from multi-dimensional data and has already been applied to osteoporosis diagnosis. For example, E et al. [12] attempted to improve the low accuracy of osteoporosis prevalence predictions using machine learning, and Kim et al. [13] applied machine learning techniques to pre-screen osteoporosis in postmenopausal women in Korea.

Feature selection is important for machine learning efficiency and accuracy [14]. In most machine learning osteoporosis diagnosis methods, selected features known to influence osteoporosis are used to train machine learning models from prepared datasets [12,13,15,16]. This study instead applies a method of selecting features optimized for machine learning from high-dimensional data, rather than by filtering features in advance based on expert knowledge. The performance of this method turns out to be better than those of extant feature selection methods.

The Korea National Health and Nutrition Examination Survey (KNHANES) is a nationwide survey of Korean residents that collects general health and nutrition data, including those of bone densitometry. Lee and Lee [15], Shim et al. [16], and Yoo et al. [17] studied machine learning models that predict osteoporosis based on the features related to osteoporosis, achieving area under the receiver operating characteristic (AUC) curve performances of 0.710, 0.743, and 0.827, respectively. Based on this, the current study trains and evaluates a machine learning model that predicts osteoporosis in postmenopausal women using raw data from the KNHANES (2008–2011).

## 2. Materials and Methods

### 2.1. Data Collection

The KNHANES database was established to identify the health and nutritional statuses of Korean citizens following the 1998 enactment of Article 16 of the National Health Promotion Act. Hence, the survey has been conducted yearly with raw data released online. KNHANES include data from common participant information, health behavior surveys, health examinations, and nutrition surveys [18]. The current study uses raw data from the V-4 (2008–2009) and V-5 (2010–2011) surveys, when osteoporosis tests were performed using DXA equipment [19]. The current study’s use of KNHANES data received ethical approval from the Institutional Review Board of the Korea Centers for Disease Control and Prevention (IRB Num. IS19EISI0063). Data were downloaded from the KNHANES website (https://knhanes.kdca.go.kr/knhanes/main.do (accessed on 7 October 2020)).

### 2.2. Study Participants

Among the 21,303 participants, four exclusion criteria were applied to meet the purpose of this study. First, those who were not tested for osteoporosis were excluded. Second, osteopenia patients were excluded because the purpose of this study is to determine whether patients have osteoporosis or normal as binary classification. Third, men were excluded, as the focus of the study is postmenopausal women. Furthermore, patients who had experienced both menopause and a hysterectomy were included as they have a high chance of contracting osteoporosis, based on previous studies [20]. Fourth, only participants aged 40–69 years were included as most over 70 have already suffered osteoporosis. Considering all four criteria, 1431 participants remained.

### 2.3. Bone Mineral Density and T-Score

Osteoporosis is normally diagnosed using BMD tests via DXA, which measures the inorganic content in bone to determine the risk of fracture and identify the prevalence of osteoporosis. BMDs of the lumbar spine and femur are usually measured, but those of the wrist, finger, or heel may be substituted [21].

In KNHANES V-4 and V-5, BMD was measured in three areas: lumbar spine, femur neck, and the whole femur. The individually measured BMDs were used to diagnose osteoporosis after calculating T-scores and comparing them to the BMDs of healthy adults, according to a recommendation by the World Health Organization. The largest BMD dataset from Asia Japan (DISCOVERY-W; fan-beam densitometer, Hologic, Inc., USA) was used as the healthy adult group [22].
T-score = (BMD − group’s BMD mean) ∗ group’s BMD stddev,(1)

T-scores calculated using the above equation were classified: a T-score of −1 or higher was classified as normal, −1 to −2.5 was classified as osteopenia, and less than −2.5 was classified as osteoporosis [23]. Based on above criteria, T-scores were classified into three different classes, and only two classes (normal and osteoporosis) as a dependent variable were used to train binary classification model.

### 2.4. Experimental Design

Figure 1 displays a flowchart explaining the study design. Prior to analysis, participants were selected considering the criteria explained in Section 2.2, and data preprocessing was conducted afterward (see Section 2.5). Machine learning algorithms were applied to predict the occurrence of osteoporosis based on 20 features having high classification influence, chosen via discussions with medical specialists and feature importance scoring from trained machine learning algorithms. The features consisted of 10 biochemical screening results (Model A) and 10 survey results (Model B). When combined, Model C is obtained.

### 2.5. Data Preprocessing

Data preprocessing was performed using Python V.3.8 using the pandas, numpy, and scikit-learn libraries. Outliers and non-responses were converted into missing values (“N/A”) based on the KNHANES data guidelines, and multinominal data were analyzed using the one-hot encoding method. Feature engineering was used to integrate features with overlapping meanings by year, and all data were conditions with standard scaling prior to training.

### 2.6. Feature Selection

As shown in Figure 1, feature selection was performed based on feature importance and recursive feature elimination (RFE), following data processing. Feature importance refers a measure of the individual contribution of the corresponding feature for a particular classifier, regardless of the shape or direction of the feature effect [24]. The higher the feature importance, the greater the influence on algorithmic decision-making. RFE is a backward feature selection technique that removes features with low importance, considering the size of the input feature set. The machine learning model was trained on all features initially, and unimportant features were then eliminated from the set.

### 2.7. Machine Learning Algorithms

A total of eight different machine learning models (KNN, Decision Tree, LDA, QDA, SVC, Random Forest, AdaBoost, and Gradient Boosting Machine) were trained and evaluated based on the KNHANES data during study. Among them, three machine learning models (Random Forest, AdaBoost, and Gradient Boosting Machine) were selected with the highest performance. In this study, three ensemble machine learning algorithms were used to analyze KNHANES data: Random Forest (RF), AdaBoost, and Gradient Boosting Machine (GBM). Ensemble learning connects several weak learning algorithms to obtain stronger results, which is effective in solving classification and regression problems. RF generates a strong decision tree by combining the outputs of several randomly generated ones [25]. AdaBoost is a classification-based model that synthesizes a classifier strengthened through weight modification by combining many weak classifiers. GBM sequentially generates trees in a manner that mitigates the errors of previous trees using gradient boosting classifiers [26].

### 2.8. Model Training

The k-fold cross-validation method was used for machine learning training and verification k times by allocating verification data differently for each iteration after dividing the dataset into k folds [27]. In this study, the training and testing datasets were divided 80:20 for learning and performance measurement, and k was set to five. This study repeated this cross-validation method 10 times, followed by an accuracy comparative analysis of 50 total learned models. During training, hyperparameters were optimized using the grid-search approach, a tuning technique that computes the optimal combination of hyperparameters by verifying the performance of all possible combinations using cross-validation [28].

### 2.9. Model Evaluation

Two indicators are normally required to evaluate machine learning performance. The first is the area under the curve (AUC) score from the receiver operating characteristic curve, which is curve-plotting sensitivity vs. one minus specificity. In statistic fields, the accuracy of the machine learning model will improve as the AUC approaches one [29].

Principal component analysis (PCA) is a multivariate analysis method that finds the main components represented by a linear combination of variables by identifying the variation–covariant relationships between large quantitative variables. PCA was used in the present study to visualize the clusters of target patients using two-dimensional reduced principal component variables.

### 2.10. Statistical Analysis

As the dependent variable of this study is the T-score, point-biserial correlation and phi correlation analyses were performed to calculate correlations instead of using the Pearson coefficient. Point-biserial correlation measures the correlation when one variable is a binary variable and the other is continuous [30]. The phi correlation analysis determines the degree of correlation between two variables when both independent and dependent variables are binary [31].

## 3. Results

### 3.1. Draft Model Building

Training with 1151 features (original data), the AdaBoost model showed the best performance in terms of the AUC (0.91), followed by the GBM (0.90) and RF (0.86). See Figure 2A. Additionally, the osteoporosis per se (dependent variable) was not clearly classified into two separate groups (normal and osteoporosis) based on only two main features (PC1 and PC2), whereas the PCA was performed on 1151 features (Figure 2B).

### 3.2. Feature Selection and Statistical Analysis

Survey data are questions that the patient can directly respond to and are related to people’s life patterns. Checkup data are collected with the biochemical screening result of participants. Table 1 shows the descriptive statistics of the 20 features selected for importance, and Table 2 presents a list of 20 variables selected by referring to the feature importance as well as one-to-one correlation coefficients between each variable and DX_OST (dependent variable). As a result of the point-biserial correlation analysis, the age variable had the highest correlation at 0.540 in the positive direction, followed by age of menarche (0.24) and use of estrogen (0.17). Among the survey data, education level had the greatest negative correlation at −0.34. Serum alkaline phosphatase level was the highest at 0.233 for screening questions with a positive correlation. From the screening questions, weight (HE_wt) scored the highest negative correlation (−0.43) with the DX_OST, followed by height (HE_ht) at −0.37.

### 3.3. Models (A, B, and C) Performance

The three machine learning models were each trained using Models A, B, and C, and grid search and five-fold cross-validation techniques were used to determine the optimized hyperparameters for the best performance. The performance of Model C (Figure 3) had a high average AUC of 0.88. Models A and B had AUCs exceeding 0.80 and 0.83, respectively.

Figure 4 and Appendix A show the performance of Model C, and using the same process, the results of the best model performances of Models A and B can be viewed in the Appendix A. Figure 4A shows the result of the ROC curves for RF, AdaBoost, and GBM. The AUCs of the RF and AdaBoost algorithms were both 0.92, and the GBM showed no significant difference at 0.91. Referring to Appendix A, the performance indicators of accuracy, precision, and recall resulted in low variations among algorithms and were stable. Figure 4B shows the results of the two-dimensional PCA for the 20 selected features. Osteoporosis and normal clusters were not completely separated, but two clusters of PC1 could be distinguished between zero and one along the *x*-axis.

## 4. Discussion

In this study, a PCA was performed prior to feature selection and afterward to confirm the relationships between osteoporosis and the selected features. According to the PCA plot (Figure 2B) of the draft model, the normal and osteoporosis groups could not be clearly distinguished based on the two principal components. However, the results of Model C (Figure 4B) implied that the clusters were distinguishable on the right side (normal) and the left side (osteoporosis) based on the specific value at the *x*-axis (between zero and one). There was no significant difference between the AUC of the draft model and Model C. For Model C, the variation in the AUCs among the three machine learning algorithms was small. Thus, training machine learning models with a small number of features is more effective than using all features in terms of model efficiency and stability.

Checkup data would be used to predict the occurrence of osteoporosis with an 80% accuracy when applying Model A and survey data would be used to predict the occurrence of osteoporosis with an 85% accuracy if Model B was applied. Finally, if both checkup and survey data were available, Model C would be appropriate to predict the occurrence of osteoporosis with an 88% accuracy. To sum up, each model would be used practically depending on the type of data collected.

Prior studies selected features that potentially affected osteoporosis based on knowledge of the medical domain. However, in this study, the feature selection step was performed using medical domain knowledge alongside feature importance and RFE techniques. Instead of collecting commonly known significant features, a large dataset was used to describe the participants in as much detail as possible. A manual data preprocessing step was also necessary to improve training and prediction accuracy. For example, the beginning age of drinking is meaningful only within the group that already had experience in drinking. Therefore, in this case, a feature engineering technique was used to convert the variable into a new one combining drinking experience and the beginning age of drinking. Using this feature selection method with preprocessing, the machine learning models of this study had better results, with AUCs of 0.919 (RF), 0.921 (AdaBoost), and 0.908 (GBM). These scores are approximately 0.1–0.2 higher than the scores of the best model performance from previous studies. Although this study did not fully consider the clinical knowledge, the unique feature selection method and data preprocessing step had a positive influence on model performance via the selection of more suitable features and the merging of various raw data into more meaningful data. In particular, the features selected in this study could be classified into two different groups (i.e., checkup and survey), each of which results in an AUC score of at least 0.80. Therefore, the trained machine learning model from this study may serve as an osteoporosis assistant diagnostic program that predicts the occurrence of osteoporosis and determines the necessity of more thorough examinations.

There were several limitations in this study. First, the raw data from the KNHANES were gathered from cross-sectional observational studies performed at limited points in time across a limited sample population. Second, as participant selection was restricted to women between 40 and 69 years old, it may be difficult to generalize the results of this study to all populations in Korea.

The results of this study can be used as an auxiliary diagnosis program for osteoporosis in the future. In a further study, the models will verify if clinical data with the same features collected from medical institutions can be generalized. Furthermore, as medical image data and deep learning technology can be used for osteoporosis diagnosis, combined with the results of this study, it might be used as a more objective and accurate osteoporosis auxiliary diagnostic tool [32,33].

## 5. Conclusions

This study generated a prediction model for classifying the osteoporosis using three machine learning algorithms based on 20 features obtained through the feature selection step. The model (Model C) including both checkup and survey features, had an AUROC value (0.92) based on 20 features. Additionally, the model (Model A) with only checkup features, scored an AUROC value (0.81), and the model (Model B) with only survey features, attained an AUROC value (0.85). The trained osteoporosis prediction models when each dataset is available are expected to be useful as an auxiliary diagnostic tool for women after menopause.

## Figures and Tables

**Figure 1 healthcare-10-01107-f001:**
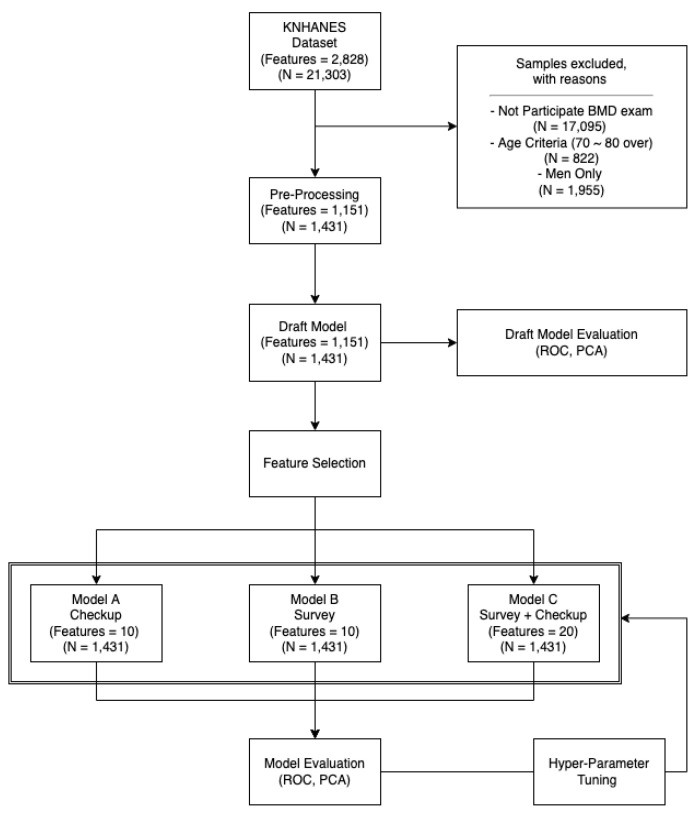
Study procedure. Model A—trained Model based on checkup features. Model B—trained Model based on survey features. Model C—trained Model based on total (checkup + survey) features.

**Figure 2 healthcare-10-01107-f002:**
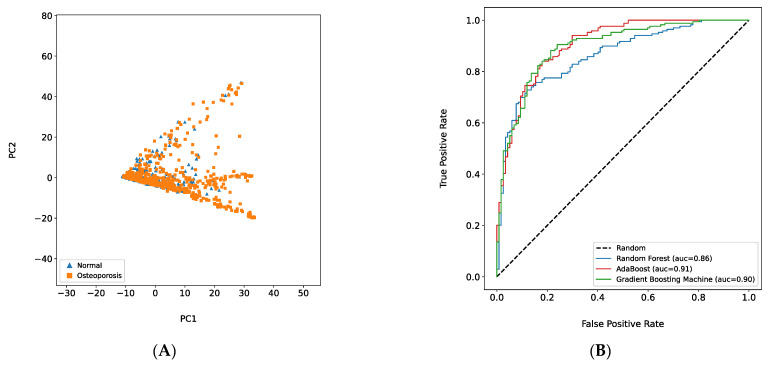
Draft Model Performance. (**A**): The result of principal component analysis plot based on 1151 features. GBM—Gradient Boosting Machine. (**B**): Receiver operating characteristic (ROC) curve for three different best models (Random Forest, AdaBoost, and Gradient Boosting Machine) based on total features (the number of features = 1151).

**Figure 3 healthcare-10-01107-f003:**
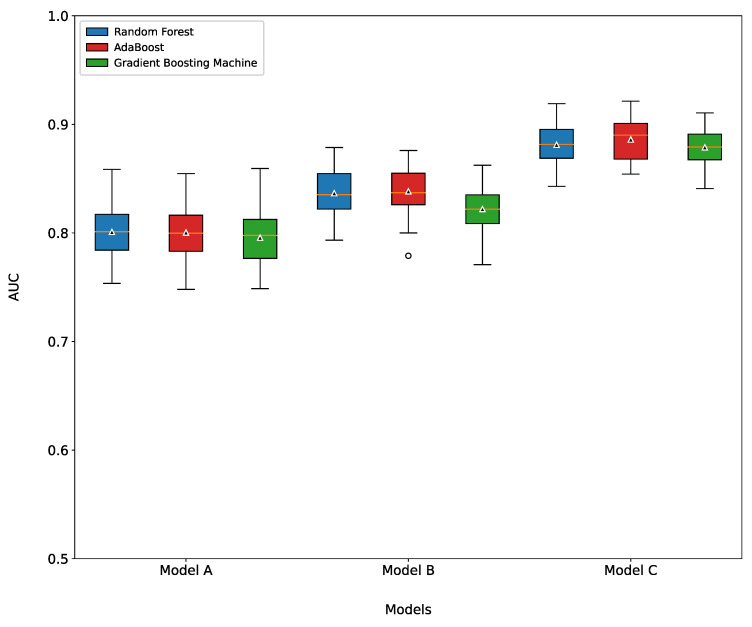
The box plot for AUROC score of three different prediction models among three different data types. Model A—trained model based on checkup features. Model B—trained model based on survey features. Model C—trained model based on total (survey + checkup) features.

**Figure 4 healthcare-10-01107-f004:**
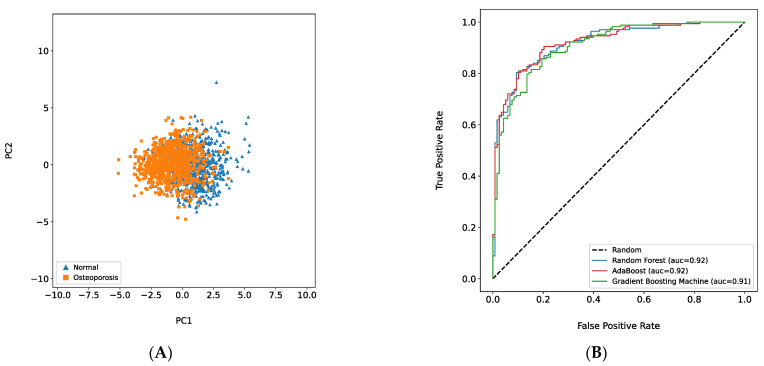
Best Model (Model C) Performance. (**A**): The result of 2D principal component analysis plot based on selected 20 features. (**B**): Receiver operating characteristic (ROC) curve of three different best models (Random Forest, AdaBoost, and Gradient Boosting Machine) based on 20 selected features (total).

**Table 1 healthcare-10-01107-t001:** Descriptive statistics of normal and osteoporosis subjects in the study.

Variables	Characteristics	Normal (*n* = 610)	Osteoporosis (*n* = 821)
Age	Age (years)	55.15 (49.46, 60.84)	62.34 (56.92, 67.77)
LW_mp_a	Age of menopause (years)	49.53 (45.07, 53.99)	48.86 (43.93, 53.78)
LW_ms_a	Age of menarche (years)	15.22 (13.37, 17.07)	16.21 (14.16, 18.26)
BP8	Average sleeping time for a day (hours)	6.6 (5.25, 7.96)	6.5 (4.93, 8.08)
BD2	Beginning age of drinking (years)	23.42 (7.69, 39.16)	22.07 (2.02, 42.12)
HE_fev1fvc	Expired lung vol. for 1	0.8 (0.75, 0.86)	0.79 (0.72, 0.86)
HE_HDL_st2	HDL cholesterol	49.58 (38.28, 60.87)	48.35 (37.58, 59.11)
HE_ht	Height (cm)	156.71 (151.59, 161.82)	152.65 (147.57, 157.72)
DX_Q_ht	Highest height of the young (cm)	158.88 (154.2, 163.56)	156.26 (151.22, 161.3)
HE_insulin	Insulin	10.7 (2.73, 18.66)	10.07 (4.5, 15.64)
LQ_VAS	Quality of life scale (index)	72.96 (54.39, 91.53)	68.32 (47.18, 89.46)
HE_ALP	Serum alkaline phosphatase (IU/L)	231.77 (165.21, 298.33)	267.75 (188.13, 347.37)
HE_sbp2	Systolic blood pressure (mmHg)	124.67 (106.22, 143.12)	127.36 (109.16, 145.56)
HE_crea	Serum Creatinine (mg/dL)	0.72 (0.62, 0.82)	0.7 (0.52, 0.89)
HE_vitD	Vitamin D (ng/mL)	18.58 (11.98, 25.18)	18.49 (11.38, 25.61)
HE_wt	Weight (kg)	62.03 (53.58, 70.48)	54.52 (47.07, 61.98)
HE_wc	Waist Circumference (cm)	83.71 (74.44, 92.98)	80.62 (71.98, 89.26)
BE5_1	Muscle exercise per week (%) *		
1	Never	80	88.94
2	One day a week	3.97	1.84
3	Two days a week	4.13	2.21
4	Three days a week	4.63	2.83
5	Four days a week	2.15	1.6
6	More than five days a week	5.12	2.58
edu	Education Level (%) *		
1	Primary or less	37.25	72.52
2	Middle	23.18	12.52
3	High	28.64	12.15
4	College or more	10.93	2.82
LW_wh	Use of estrogen (%) *		
0	No	25.96	12.93
1	Yes	74.04	87.07

* indicates categorical variables, and the number of each characteristic under categorical variables refers to percentage.

**Table 2 healthcare-10-01107-t002:** The results of univariate correlation analysis with the list of 20 independent variables and dependent variable.

Data Type	Variables	Characteristics	Correlation
Checkup	HE_wt	Weight (kg)	−0.426 (−0.467, −0.383)
	HE_ht	Height (cm)	−0.367 (−0.411, −0.321)
	HE_wc	Waist Circumference (cm)	−0.170 (−0.219, −0.119)
	HE_fev1fvc	Expired lung vol. for 1 s	−0.115 (−0.172, −0.056)
	HE_HDL_st2	HDL cholesterol (mg/dL)	−0.055 (−0.108, −0.002)
	HE_insulin	Insulin (μIU/mL)	−0.046 (−0.103, 0.010)
	HE_Crea	Serum Creatinine (mg/dL)	−0.045 (−0.098, 0.008)
	HE_vitD	Vitamin D (ng/mL)	−0.006 (−0.059, 0.047)
	HE_sbp2	Systolic blood pressure (mmHg)	0.073 (0.021, 0.124)
	HE_ALP	Serum alkaline phosphatase (IU/L)	0.233 (0.183, 0.283)
Survey	Edu	Education Level	−0.345 (−0.390, −0.298)
	DX_Q_ht	Highest height of the young (cm)	−0.261 (−0.317, −0.203)
	LQ_VAS	Quality of life scale (index)	−0.112 (−0.163, −0.060)
	BE5_1	muscle exercise per week (days)	−0.107 (−0.158, −0.055)
	LW_mp_a	Age of menopause (years)	−0.070 (−0.123, −0.016)
	BD2	Beginning age of drinking (hours)	−0.037 (−0.088, 0.015)
	BP8	Average sleeping time for a day (years)	−0.034 (−0.085, 0.018)
	LW_wh	Use of estrogen	0.17
	LW_ms_a	Age of menarche (years)	0.243 (0.192, 0.292)
	Age	Age (years)	0.540 (0.503, 0.576)

Parentheses under the correlation column indicate a 95% confidence interval.

## Data Availability

The data presented in this study are available to download at https://knhanes.kdca.go.kr/knhanes/main.do (accessed on 7 October 2020) for research purposes. It is necessary to obtain IRB registration of the KNHANES dataset for publishing the study.

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
