# Peer review of "Osteoporosis Pre-Screening Using Ensemble Machine Learning in Postmenopausal Korean Women"

_healthcare, 2022, doi:10.3390/healthcare10061107_

Round 1

Reviewer 1 Report

In my opinion, the quality of the article is high and the discussed solution has great application potential. The paper writing is professional and easy to understand. However, I have the following concerns which are necessary to be addressed in the revision to further improve the quality of the manuscript.

Comments:

1.    Some more recent and relevant papers could be cited to support the literature review part of the paper.

2.    The quality of the figures is low. Need revision.

3.    Further clarification on the future work is needed.

Author Response

Comment: Some more recent and relevant papers could be cited to support the literature review part of the paper.

Answer: Agree with your comment. So, four relevant papers were added to support the literature review part of the manuscript (Line 32, Line 33, and Line 35).

Comment: The quality of the figures is low. Need revision.

Answer: According to the comment, we changed the figures to 500 dpi.

Comment: Further clarification on the future work is needed.

Answer: Further clarification on the future work is added at the end of the discussion section. (Line 292 ~ 297)

I really appreciate your interest in my manuscript. Thank you for your all comments. I did my best to answer all the comments. 

Reviewer 2 Report

The comments are attached 

Author Response

Thank you for taking the time to review and comment on this study. I tried my best to answer all the comments from reviewer. 

Reviewer 3 Report

The manuscript used data from KNHANES to predict a patient’s risk of osteoporosis through using three machine learning algorithms to train three models. The model C achieved excellent performance based on the unique feature selection method and data preprocessing step, which may be utilized to predict the occurrence of osteoporosis. Therefore, I would like to recommend the paper to be accepted after addressing the following issues:

1.               Random forest, AdaBoost, and gradient boosting machine learning algorithms were used in this manuscript. But the authors did not explain why the three machine learning algorithms are chosen in the manuscript.

2.               The features selected in this study could be classified into two different groups: checkup and survey. The basis of this classification is not clear in the manuscript.

3.               The figure caption and figure content could not match in Figure 2 and Figure 4. Please correct it.

4.               In Figure 3, the AUROC scores of the three models based on the gradient boosting machine learning algorithm are smaller than the other two machine learning algorithms. Please give the explanation in the manuscript.

Author Response

Comment: Random forest, AdaBoost, and gradient boosting machine learning algorithms were used in this manuscript. But the authors did not explain why the three machine learning algorithms are chosen in the manuscript.

Answer: I agree with your comments. I added an extra explanation on the 2.7 Feature selection part in material & method. (Line 135 ~ Line 141).

Comment: The features selected in this study could be classified into two different groups: checkup and survey. The basis of this classification is not clear in the manuscript.

Answer: Survey data are questions that the patient can directly respond to and are related to people's life patterns. Checkup data are collected with the biochemical screening results of participants. The above information is added to the manuscript (Line 194 ~ 196).

Comment: The figure caption and figure content could not match in Figure 2 and Figure 4. Please correct it.

Answer: The figure 2 and 4 captions and content were corrected now. Thanks for your comments.

Comment: In Figure 3, the AUROC scores of the three models based on the gradient boosting machine learning algorithm are smaller than the other two machine learning algorithms. Please give the explanation in the manuscript.

Answer: I agree with your comment. However, the y-axis range in Figure 3 is considered to magnify between 0.750 and 0.925. In fact, the GBM model is only 0.1 percent different from Adaboost and Random Forest. In order to eliminate misunderstandings, we consider changing the range of the y-axis between 0.5 and 1.0. 

Thank you so much for your comments. I did my best to answer all the comments that you made. 

Round 2

Reviewer 2 Report

I read the answers of the authors with care, and I would like to say that the definition and the diagnostic criteria as issued by the WHO is superficial and suits only general practitioners. Experts in bone disorders can never adopt the principles of the WHO. 

Anyhow, authors need to add the following  to line 275: This paper lacks the required clinical knowledge.  In this case, I accept publication. Best regards

Author Response

Thank you for your comments. I carefully considered and decided to change lines 274 ~ 276. I really appreciate your thoughtful comments. 
